# Noise Indicators Relating to Non-Auditory Health Effects in Children—A Systematic Literature Review

**DOI:** 10.3390/ijerph192315633

**Published:** 2022-11-24

**Authors:** Michail Evangelos Terzakis, Maud Dohmen, Irene van Kamp, Maarten Hornikx

**Affiliations:** 1Building Acoustics Group, Department of the Built Environment, Eindhoven University of Technology, 5612 AZ Eindhoven, The Netherlands; 2National Institute for Public Health and the Environment, 3721 MA Bilthoven, The Netherlands

**Keywords:** noise indicators, noise exposure, environmental noise, non-auditory health-effects, children

## Abstract

A systematic literature review was conducted to investigate which objective noise indicators related to various noise sources (i.e., aircraft, road-traffic, and ambient noise) are the best predictors of non-auditory health-effects in children. These relationships are discussed via a conceptual framework, taking into account main parameters such as the type of noise source, the exposure locations and their environments, the type of noise indicators, the children’s mediating factors, and the type of non-auditory health effects. In terms of the procedure, four literature databases were screened and data was extracted on study design, types of noise sources, assessment method, health-based outcomes and confounders, as well as their associations. The quality of the studies was also assessed. The inclusion criteria focused on both indoor and outdoor environments in educational buildings and dwellings, considering that children spend most of their time there. From the 3337 uniquely collected articles, 36 articles were included in this review based on the defined inclusion and exclusion criteria. From the included literature, it was seen that noise exposure, assessed by energetic indicators, has significant associations with non-auditory health effects: psychophysiological, cognitive development, mental health and sleep effects. Percentile and event-based indicators provided significant associations to cognitive performance tasks and well-being dimension aspects.

## 1. Introduction

Urban noise is one of the modern health threats for human beings [1]. Like the elderly, children can be considered a sensitive group of people [2]. Because of their continuous physical and mental development, noise exposure may cause detrimental and irreversible health-related effects later on in their lives [3]. Auditory health effects (e.g., hearing loss and tinnitus) are considered as direct consequences of noise exposure, corresponding to the injury of the auditory system [4]. Usually, environmental noise exposure does not reach harmful levels for traumatizing children’s hearing [3]. However, noise exposure has been characterized as a risk factor for the presence of tinnitus in youth [5]. Noise, as a nonspecific stressor, is also responsible for the presence of health effects as a result of chronic noise exposure [4]. The relation between non-auditory health effects and noise exposure in children has been investigated, including psychophysiological [6,7,8,9,10,11], cognitive [7,9,10,12,13], mental [14,15], sleep [16,17], as well as physical development aspects [18,19,20,21]. In contrast to adults, children are more sensitive to noise stimuli, indicating that coping repertoires may have not been entirely developed [22].

Various mechanisms have been proposed for the relationship between noise exposure and health effects in humans (e.g., [4,23,24,25,26]). Aspects, such as those related to noise exposure (e.g., type of noise source) and its characteristics (e.g., duration and temporal variations), as well as to human organism (e.g., stress reactions) are mainly considered. Furthermore, mediating factors, associated to personal appraisal and coping repertoires, are also included [24,25]. These correspond to the assessment of noise exposure as threat or no threat as well as to the presence of physiological reactions with respect to individuals acquired and genetic characteristics [24]. Noise-induced health effects have been divided into two main categories; the short-term and long-term effects. Short-term effects are associated to disturbance of intended activities (e.g., concentration, sleep, performance), annoyance, and stress responses [4,23,24,25], whereas psychophysiological reactions (e.g., cardiovascular, and neuroendocrine aspects), and disorders (e.g., cardiovascular, and mental disorders) are identified as long-term health effects [24].

In contrast to some other environmental factors (e.g., benzene exposure), which their components can be accumulated in organisms, noise exposure is not an exogenous factor [27]. Hence, its effects can only be estimated based on the characteristics of noise exposure through objective noise indicators [27,28]. For the quantification of the noise exposure, various types of noise indicators have been used. The most commonly used noise indicators are the energetic, statistical, and the event-based indicators, weighted by the *A*-filter. In general, the (energetic) *A*-weighted equivalent noise indicator, $L_{A,eq,T}$, is the most commonly used indicator in noise exposure studies. This indicator is defined as the equivalent sound level over a specific period of the time, *T*. The day-evening-night equivalent sound level ($L_{den}$) indicator is calculated using $L_{A,eq,T}$, adding a penalty factor of 5dB for the evening noise levels, from 19:00 to 23:00, and of 10 dB for the night noise level, from 23:00 to 07:00 [4]. The additional penalty factors express the increase in sensitivity with respect to the restoration and the sleep period [23]. Sub-divisions of $L_{den}$ into, day ($L_{d}$), evening ($L_{e}$), night ($L_{n}$), day–evening ($L_{de}$), evening–night ($L_{en}$), and day–night ($L_{dn}$) are also used in noise exposure studies.

Percentile (statistical) indicators are used for capturing both noise events and noise floors. The *A*-weighted percentile level indicators ($L_{AN}$) are used for estimating the noise levels exceeded for N% of the measurement time. The most commonly used percentiles are the 1st, 5th, 10th, 50th, 90th, 95th, and 99th. The (median-based) $L_{A50}$ indicator is correlated to the average perception, since it is not affected by extraneous temporary noise events [29,30]. The $L_{A01}$ indicator is used for the identification of the most dominant noise levels. The $L_{A05}$ and $L_{A10}$ are used for the characterization of the intermittent/intrusive noise levels, while the $L_{A90}$ and $L_{A95}$ are used for the description of background noise levels, respectively. Furthermore, the $L_{A99}$ indicator is used for the estimation of the underlying noise levels [31]. Combinations of indicators, such as $L_{A05}$–$L_{A95}$ and $L_{A10}$–$L_{A90}$, are commonly used for describing noise variation with respect to noise levels [32]. The sound exposure level (SEL) indicator is similar to $L_{A,eq}$. However, instead of averaging the total sound energy over the measurement period, a reference duration of one second is used. This indicator is useful when multiple aircraft events need to be compared [33]. Both $L_{A,max}$ and $L_{A,min}$ are capable of identifying both the maximum level of noise events and the noise floor over the time-period *T*.

Other types of indicators have been developed for the discrimination of the sound environments, related to noise variation, spectrum and noise emergence. However, these indicators have not been used to investigate possible relationships between noise exposure and noise-induced health effects in children [32].

The aim of this systematic literature review is to assess the objective noise indicators associated to non-auditory health effects in children. Children’s appraisal and coping strategies are continuously and rapidly developed during the childhood period. Therefore, it is important to identify the objective noise indicators associated with non-auditory health effects in children with respect to the type of the noise source, the locations of noise assessment, the exposure locations and their environments, the age of children, as well as parameters that influence the identified relationships. To our knowledge, this is the first review focusing on the association between noise indicators and non-auditory health effects in children. Factors that influence noise exposure are also discussed.

This paper is structured as follows; Section 2 provides the procedure of the literature review. Section 3 presents the results extracted by the included literature. Section 4 discusses the results with respect to a conceptual framework. Section 5 and Section 6 pinpoint the limitations as well as a the main remarks of this work, respectively. Finally, in the Appendix A, a detailed overview of the used literature is given.

## 2. Procedure

In this section, the search strategy and procedures for identifying and assessing the collected literature, with respect to the defined inclusion and exclusion criteria, are presented.

### 2.1. Search and Screening Strategy

A systematic literature review enables the improvement of decisions made from concluded literature [34], minimizing the risk of bias occurred by a subset of literature.

In this systematic literature review, three key terms were defined; (1) noise exposure, (2) health effects and (3) children. In this way, relevant publications were collected. Both key and search terms are summarized in Table 1.

As shown in Table 1, only noise-based search terms were used, and no sound- or acoustics-related terms. The {Health Effects} key term focuses on the non-auditory health effects, including psychophysiological, cognitive, mental health, and sleep effects. Finally, the {Children} key term covers various developmental stages.

### 2.2. Inclusion and Exclusion Criteria

Literature from 2000 until 2020 is considered in this review. This is motivated by the high number of studies conducted after the year 2000. In terms of noise indicators, only objective noise indicators related to non-auditory health effects in children-based studies are considered. Studies based on subjective noise indicators (e.g., noise logs), corresponding to the individuals’ perception of noise, were excluded. Regarding the type of noise exposure, ambient, aircraft, and road traffic noise are selected. Ambient noise exposure corresponds to the inclusion of all noise sources in an environment in which road-traffic noise is the dominant one. Dwellings and educational buildings are considered as the main locations, since children spend most of their time in these settings. In addition to this, both indoor and outdoor environments are considered. Indoor environments may provide either a significant reduction of outdoor noise exposure or the presence of secondary indoor noise sources (e.g., irrelevant speech in classrooms). Finally, studies that do not include noise exposure in the statistical analyses as well as studies using reproduced sound levels played back by loudspeakers or headphones were also excluded. In Table 2, both the inclusion and exclusion criteria are summarized.

### 2.3. Search Procedure

The search procedure was conducted in four different literature databases (ScienceDirect, Scopus, MDPI, and PubMed) in August 2020 and the collected literature was stored into the freeware software, Zotero (https://www.zotero.org/ (accessed on: 25 November 2020)), version 5.0.93. In the search procedure, the databases were searched based on the presence of the defined search terms (see Table 1) in the title and/or the abstract, and/or the full text of the articles. In the search query, the key terms were separated, using the *AND* operator, and the search terms were separated, using the *OR* operator. The full searching query is included in Appendix A.

### 2.4. Screening Procedure

In this systematic literature review, the PRISMA (http://prisma-statement.org/ (accessed on: 17 November 2020)) approach was followed (see Figure 1). After collecting the literature from the four databases (n = 8766), the screening of the identified literature was conducted. This includes the identification of duplicate articles in the collected literature. During this phase, the articles were also screened in terms of the criteria (A), (B), and (C) via the automatic function and built-in search filters in Zotero software (n = 3337). Then, the three-stage screening procedure (i.e., title, abstract and full-text screening) was conducted based on the defined inclusion and exclusion criteria.

In the first screening stage (title screening), the identification of key words, related to the inclusion and exclusion criteria (D)–(I), in the title of each article was conducted from the first author (M.E.T.). Papers with non-informative titles were passed to the next stage. From this stage, the literature was narrowed down to 1056 articles. In the second screening stage (abstract screening), the same procedure, as in the title screening, was followed, reducing the literature further to 224 articles. This stage was conducted by the first author (M.E.T.). When the abstracts did not provide sufficient information to judge, the conclusion sections were then evaluated. Again, when the abstracts and the conclusions of the papers were not informative, these papers were passed to the final stage. The third screening stage (full-text screening) corresponds to the reading of the remaining articles with respect to the defined criteria (D)–(I). This round was conducted in two steps. In the first step, the reading of the articles was performed by the first author (M.E.T.), narrowing down the literature to 35 articles. In the second step, these articles were read by the second author (M.D.), validating that these articles fulfill the defined criteria (i.e., eligibility assessment). During this procedure there was no deviation between both authors (M.E.T. and M.D.) in terms of the inclusion of 35 articles to the included literature. Furthermore, the second author (M.D.) validated that both the data and the outcomes per each article were extracted in a correct way.

Finally, although the searching procedure is conducted in four databases, important articles may be missing out. Hence, additional articles may be imported to the concluded literature, after the full-text screening procedure, through a manual insertion. In that way, the systematic approach was not violated. In this work, only a single article was added in the concluded literature, which was in first author’s (M.E.T.) knowledge, increasing the included literature to 36 (single) articles.

### 2.5. Criteria for Assessing the Risk of Bias

The quality (risk of bias) of each included article in this review was assessed with respect to the criteria in the Table 3. These criteria were developed by WHO [35] and lately further re-developed by van Kempen et al. [36]. These criteria have been adjusted to the needs of this systematic literature review. None of the original defined criteria was excluded. However, an additional criterion was added, assessing the short-term noise exposure. Overall, high quality of evidence is associated to a low risk of bias and vice versa. The grading of the risk of bias in each article per health effect is presented in the Appendix A.

## 3. Results

In this section, the results extracted by each article of the included literature are summarized. For a more detailed overview of the results extracted by each article, the reader is directed to the Appendix A.

### 3.1. Results Overview per Health-Effect

From the included literature, information on health effects, health aspects, parameters, noise sources, and noise indicators was extracted (Table 4). This information is summarized with respect to the health effects: psychophysiological, cognitive, mental health, and sleep effects (Table 5, Table 6, Table 7, Table 8, Table 9, Table 10 and Table 11). It must be noted that some articles address multiple health effects, and therefore a doubling of articles occurs in Table 4. A detailed and extensive overview is included in Appendix A.

### 3.2. Noise-Induced Effects and Noise Indicators

In this sub-section, the main results are summarized, such as presented in Table 5, Table 6, Table 7, Table 8, Table 9, Table 10 and Table 11. Further details can be found in the Appendix A.

#### 3.2.1. Cardiovascular Effects

Cardiovascular aspects related to systolic and diastolic blood pressure were significantly associated with both aircraft and road-traffic noise. The $L_{A,eq}$ indicator presented significant associations with both types of blood pressure [38,40]. In the case of the van Kempen study [40], significant associations between the $L_{A,eq}$ indicator and the diastolic blood pressure were presented when aircraft noise and annoyance were simultaneously considered. Focusing on road-traffic noise exposure, the $L_{den}$ indicator [37,39] provided significant associations mostly with systolic and less so with diastolic blood pressure. Finally, the $L_{dn}$ indicator showed no significant relationship with any type of blood pressure [41].

#### 3.2.2. Neuroendocrine Effects

Cortisol, gluccocorticoid, and their metabolites were the main neuroendocrine aspects explored with respect to road-traffic and aircraft noise exposure. Although the $L_{den}$ indicator provided significant relationships between road-traffic noise exposure and some cortisol [43] and gluccocorticoid [43] metabolites, these associations were not seen in terms of their total levels [42,43]. Focusing on the $L_{dn}$ indicator, significant associations to cortisol and its 20a-dihydcortisol metabolite were found with respect to road-traffic noise exposure [41]. With regard to aircraft noise exposure, non-significant associations with cortisol levels relative to the SEL indicator were extracted [44].

#### 3.2.3. Cognitive Development Effects

Cognitive development tasks related to sustained attention, reading/comprehension, short-term memory (i.e., working memory) and long-term memory (i.e., recognition, recall, and prospective memory) were explored with respect to both aircraft and road-traffic noise exposure. Studies based on both types of noise exposure showed no significant associations with sustained attention with respect to the $L_{A,eq}$ indicator [47,51]. However, a significant relationship between sustained attention and exposure to aircraft noise, expressed by the SEL indicator, was identified [44].

Although no clear associations between reading/comprehension tasks and both types of noise exposures were extracted based on the $L_{A,eq}$ indicator [44,45,46,47,48,49,50,51], the more consistent ones were found in studies where the age of the children was between 7 and 11 years [44,46,47,50]. In terms of the SEL indicator-based study [49], significant associations between aircraft noise exposure and reading tasks were identified.

Regarding memory tasks, long-term memory was found to be impaired by noise exposure, when it is expressed by the $L_{A,eq}$ indicator, compared to short-term memory. More specifically, working memory tasks, associated to short-term memory, did not reveal any significant association with respect to aircraft [44,47] and road-traffic [47] noise exposure. However, recognition memory tasks, associated to long-term memory, were significantly impacted by both types of noise exposure [44,47,52]. Although both significant [51,52] and non-significant [47,48] associations were with respect to recall memory, prospective memory showed no significant effects with regard to exposure to noise from both aircraft and road-traffic [47,52].

#### 3.2.4. Cognitive Performance Effects

Noise events, expressed by the $L_{A,max}$ and the $L_{A10}$ indicator, as well as background level, expressed by the $L_{A90}$ indicator, presented significant associations to cognitive performance tasks (i.e., language, mathematics, and science), considering exposure in educational buildings [56]. Ambient exposure in both dwellings, expressed by the $L_{den}$ indicator, and educational buildings, expressed by the $L_{d}$ or the $L_{e}$ or the $L_{n}$ indicator, revealed significant associations to performance tasks (i.e., mathematics and language) [54]. However, neither the $L_{den}$ levels at dwellings nor the $L_{d}$ levels at educational buildings provided significant associations with complex cognitive tasks (e.g., mathematics) [54]. Due to aircraft noise, both language and mathematics tasks were found to be significantly affected in educational buildings, expressed by the $L_{A,eq}$ indicator [55]. Considering cognitive functioning tasks, significant associations were seen with respect to both aircraft and road-traffic noise exposure in educational buildings, expressed by the $L_{A,eq}$ indicator [40,57]. Finally, academic achievements were not impacted by road-traffic noise exposure, expressed by energetic, statistical, and event-based noise indicators [53].

#### 3.2.5. Well-Being Dimensions

Well-being dimensions related to annoyance, motivation and quality of life were investigated with respect to both aircraft and road-traffic noise exposure. Annoyance was the well-being dimension which presented strong associations with both aircraft and road traffic noise exposure, expressed by various indicators, such as the $L_{A,eq}$[44,45,46,47,58,59,60,61,63], the $L_{den}$[62], the $L_{A,max}$[63], the $L_{A90}$[63] and the SEL [49] indicator.

The impact of aircraft noise exposure on motivation was only addressed by one study. In this study [44], a non-significant relationship was found between motivation and the $L_{A,eq}$ indicator.

Finally, the well-being dimension that corresponds to the quality of life showed significant associations with both exposure to aircraft and road traffic noise with respect to the $L_{A,eq}$[46] and the $L_{dn}$ [41] indicator, respectively.

#### 3.2.6. Mental Health Effects

Mental health effects related to mental disorders, behavioral problems, executive functioning, and perceived general and mental health were investigated with respect to aircraft and road-traffic noise exposure. Mental disorders associated with anxiety and depression did not present a significant association with aircraft noise exposure, expressed by either the $L_{A,eq}$[44] or the SEL[49] indicator. Focusing on the behavioral problems, significant associations were revealed with respect to the $L_{dn}$ indicator, expressing road traffic noise exposure [68]. However, this significance was not present with respect to the externalizing or internalizing behavior problems [68]. Executive functioning skills, related to attention deficit disorders, only showed significant associations for boys in relation to road traffic noise expressed by the $L_{A,eq}$ indicator [70].

Psychological morbidity is assessed via the Strengths and Difficulties Questionnaire (SDQ), mainly including five items; hyperactivity/inattention, emotional symptoms, conduct problems, pro-social behavior and peer problems. The total SDQ scores did not provide significant associations with both aircraft [44,45,65,69] and road traffic noise [64,65,66,69] exposure, expressed by the energetic indicators $L_{A,eq}$[44,45,65,69] and $L_{den}$[64,66]. Focusing on each item separately, both significant and non-significant relationships were extracted. More specifically, the hyperactivity/inattention item presented the strongest associations with respect to the $L_{den}$ indicator and road traffic noise exposure in dwellings [64,66] as well as the $L_{A,eq}$ and aircraft noise exposure in educational buildings [65,69]. The inattention-based study [67] revealed significant associations to road traffic noise exposure and the $L_{den}$ indicator. In terms of the emotional symptoms item, this presented mostly non-significant associations with regard to both types of noise exposures, assessed by both energetic indicators, $L_{A,eq}$[44,45,65,69] and $L_{den}$[66]. Only one study [64] presented significant associations between road-traffic noise exposure levels, expressed by the $L_{den}$ indicator, and the emotional symptoms item. The conduct problems item presented a non-significant association with noise exposure in most studies [44,45,64,66]. However, the $L_{A,eq}$-based studies on aircraft and road-traffic noise exposure identified significant associations with this item [65,69]. The pro-social behavior [44,64,65,66] and peer problem [44,64,65,66] items did not reveal any significant association with regard to any type of noise exposure and noise indicator.

Finally, perceived mental and general health aspects did not reach any significant association with either aircraft or road traffic noise exposure, expressed by the $L_{A,eq}$ indicator [40]. However, a significant relationship was revealed between aircraft noise exposure, expressed by the SEL indicator, and stress symptoms [49].

#### 3.2.7. Sleep Effects

Sleep aspects such as sleep quality, waking-up episodes, sleepiness, and difficulties falling asleep were investigated with respect to road-traffic noise exposure. Sleeping problems and problems to fall asleep were significantly related to the $L_{n}$ indicator, only at the least exposed façade in dwellings, rather than at their most exposed façade [64]. Significant associations between noise exposure, expressed by the $L_{A,eq}$ indicator, and sleep parameters, related to sleep quality, wrist activity, and feeling alertness in daytime [71] were also indicated. Non-significant associations were revealed with respect to noise exposure, expressed by the $L_{en}$ indicator, and sleep quality [72].

## 4. Discussion

Based on the included literature, it is seen that noise-induced health effects have shown significant relations to various noise indicators (see Table 5, Table 6, Table 7, Table 8, Table 9, Table 10 and Table 11). Aspects such as the type of noise source, the sound propagation path, and the acoustic characteristics of the environments influence both noise exposure and its quantification. In combination with the developmental characteristics of children, which act as mediators, non-auditory noise-induced health effects may be of short-term or long-term nature. Hence, a conceptual framework is introduced, representing and discussing all the components associated to the possible noise-induced health effects in children with respect to the findings from the included literature.

### 4.1. Conceptual Framework

The conceptual framework in Figure 2 describes the relationship between noise exposure and non-auditory health effects. The main components of this framework are: (1) the type of noise source, (2) the exposure locations, (3) the exposure environments (i.e., outdoor and indoor environments), including the acoustic factors of both environments as well as of the outdoor-to-indoor infrastructure (i.e., building’s façade), (4) the assessment of noise exposure through objective noise indicators, (5) the children’s mediating factors, and (6) the types of non-auditory health effects (i.e., degree of severity).

Overall, the conceptual framework can be interpreted in the following way. Although noise sources are defined based on their own characteristics (i.e., temporal and spectra), noise exposure is associated with the type of the noise source, the location’s and its environments’ characteristics. In outdoor environments, mostly no free-field conditions are encountered. Therefore, outdoor environment characteristics, such as, the street width and height as well as the distance between building blocks, modify both the temporal and spectra acoustic characteristics of the noise sources, defining the outdoor noise exposure. Furthermore, parameters related to building acoustics (i.e., façade sound insulation) and room acoustics are responsible for the further modification of the outdoor exposure transmitted into the indoor environment. In that way, noise indicators (energetic, statistical or event-based) are the quantities for correlating noise exposure and/or its characteristics with non-auditory health effects. Finally, mediating factors are responsible for the possible strengthening of the associations between noise exposure quantification and possible health effects in children.

#### 4.1.1. Noise Source

The noise source defines the type of noise exposure based on characteristics related to its spectral and temporal components. These parameters are perceived differently by each child (i.e., suggesting that factors such as personal assessment and coping strategies are responsible for the perception of noise exposure as harmful or not harmful), influencing non-auditory health effects (e.g., degree of discomfort).

From the included literature, it was seen that 47% of the studies focused on the road-traffic noise exposure, 20% focused on the aircraft noise exposure, 22% focused on both the road-traffic and aircraft noise exposure, and the remaining 11% focused on ambient noise exposure, in which its dominant noise source is usually road-traffic noise.

Both spectral and temporal characteristics of environmental noise sources, such as cars, motorcycles, and aircraft, have changed over the years. As a typical example, the improvement in aircraft jet engines over the decades can be considered, in which the generated sound power was significantly reduced since their introduction [73]. Consequently, this indicates that spectral characteristics of noise sources have been modified. Although this is known, spectral information was not presented by any of the studies in the included literature. This indicates the importance of interpreting the associations between noise exposure and health effects in various studies with caution, especially in the case of studies with a large time gap between them.

#### 4.1.2. Locations and Environments

Considering that children spend most of their time in residential and educational buildings, noise exposure in indoor environments is of greater importance in comparison to the noise exposure in outdoor environments.

As regards the outdoor noise exposure, this depends on parameters such as the type of street (e.g., a street canyon or an open street), the width of the street, the distance to the noise sources as well as the acoustic characteristics of the outdoor environment. For the indoor environment, the characteristics of the building skin and indoor room acoustic characteristics influence the transmitted environmental noise exposure. Note that secondary noise sources from surrounding indoor environments (e.g., irrelevant speech in school environments and neighbor noise in dwellings) may either influence or even dominate the noise transmitted from outdoors. When considering the effect of outdoor noise exposure on the indoor environment, it is therefore important to ensure that secondary noise sources do not dominate the indoor noise exposure. Considering the façade characteristics, factors such as the façade material, the presence of balconies or obstacles, the type of the windows (i.e., single or double glazing), as well as the glazing area over façade area (of the room under consideration) influence the temporal and spectral characteristics of indoor noise exposure.

Looking at the locations and the environments where noise exposure was assessed (Table 5, Table 6, Table 7, Table 8, Table 9, Table 10 and Table 11), approximately 33% of the items were for homes, 47% for educational buildings, and 20% for both locations. In terms of the environments, in dwellings-based studies, all the studies considered outdoor environments. In educational buildings-based studies, approximately 31% focused on outdoor environments, 8% on indoor environments, and 8% on both indoor and outdoor environments. Studies based on both dwellings and educational buildings considered only outdoor environments.

The absence of the assessment of indoor noise exposure (i.e., not taking into account the acoustic characteristics of the building façade and indoor environment) may lead to weak relationships between indoor noise exposure, based on outdoor noise exposure, and non-auditory health effects, even though these are statistically significant.

Focusing on façade characteristics, these modify the transmitted outdoor exposure into the indoor environment. In particular, the glazing area of the windows and its ratio over the façade area can be considered as important parameters. It was found that indoor noise levels are affected by the positions of the windows (i.e., open, tilted, and closed) [74]. This is important in both sleep and cognitive studies, taking into account the effects of sleep quality on cognitive tasks [75]. According to Locher et al. [74], the equivalent noise level difference of between outdoor and indoor environments in dwellings ranges from 1.4 to 17.3 dBA, when measurements with open windows were conducted.

The aforementioned aspects can be cast into acoustic parameters. Room acoustics parameters such as the reverberation time and speech intelligibility have shown significant relations to acoustic comfort [76]. The term acoustic comfort refers to the perceptual pleasant evaluation of an environment with respect to its acoustics characteristics [77]. Outdoor acoustics characteristics, such as these related to reverberation time, modify the noise exposure, which may lead to a significant impact on acoustic comfort. Finally, building acoustics characteristics related to parameters such as façade acoustics insulation have presented strong associations to acoustic comfort in the indoor environment [78].

Although these parameters are known, only 11% of the included literature took into consideration room and building acoustics parameters for investigating their impact to well-being dimensions aspects. For example, Minichilli et al. [61] included the reverberation time (RT), the speech transmission index (STI), the airborne façade insulation ($D_{2m,nT,w}$) as well as the wall sound reduction index ($R_{w}^{{}^{\prime }}$) in classrooms. In this study, although annoyance presented only a good correlation with RT and not with any of the other aforementioned acoustics indicators, none of these acoustic factors was included in statistical analysis as adjusted factors. Reverberation time was also included by Klatte et al. [46], showing that RT was unrelated to aircraft noise exposure and children’s reading scores, excluding it from the analysis. The lack of measuring the reverberation time in a school that was relocated was another reason for excluding RT from the analyses. In terms of sound insulation, this was included in the analyses with respect to the type of window glazing as adjusting factors by Stansfeld et al. [47] and van Kempen et al. [60]. In the case of Klatte et al. [46], the acoustics insulation was estimated using a combination of variables, including fenestration, glazing and wall thickness. The need of including acoustics factors related to the indoor environments is pinpointed by both Silva et al. [59] and Stansfeld et al. [65]. It needs to be mentioned that in the acoustics factors, the frequency dependency was not considered. Finally, regarding the outdoor environment, it needs to be noted that the acoustics characteristics of the outdoor environments (e.g., reverberation time) have not been considered by any of articles in the included literature.

#### 4.1.3. Noise Indicators

Looking at Table 4, mostly energy-based noise indicators have been used in children-based studies. More specifically, 83% of the included articles assessed noise exposures via energetic noise indicators, from which 50% corresponds to the $L_{A,eq}$ indicator, 25% to the $L_{den}$ indicator, and the remaining 8% to relative subdivisions of the $L_{den}$ indicator, referring to the $L_{d}$, the $L_{e}$, the $L_{n}$, the $L_{en}$ and the $L_{dn}$ indicator. Short-term energetic noise indicators were also used, quantifying typical noise levels in focused spaces and time periods (e.g., during task periods). Although that energetic noise indicators are associated to long-term health effects, these hide information related to noise levels’ dynamics and temporal characteristics [32]. Hence, these factors may provide weak association even when the relationships between noise exposure and non-auditory health effects are statistically significant.

A lower percentage of the collected literature included statistic percentile-based indicators (e.g., $L_{A10}$ and $L_{A90}$) and event-based indicators (e.g., $L_{A,max}$ and SEL). These indicators cover 17% of the included literature for investigating mostly the relationship between noise exposure and cognitive performance tasks. By considering that the $L_{A10}$ and the $L_{A90}$ indicator describe the intermittent/intrusive levels and the background noise level, respectively, these indicators could provide important information in terms of the influence of the dominant noise events and background noise on health effects. Moreover, the $L_{A,max}$ indicator can provide similar information to the percentile-based indicator $L_{A01}$. According to Secchi et al. [79], the $L_{A,max}$ is characterized by a high dispersion of data in exposure levels in comparison to the $L_{A01}$ under a defined time interval length [79]. Therefore, it is recommended to use the $L_{A01}$ index instead of the $L_{A,max}$ index to investigate the effects of noise exposure on cognitive performance. In terms of noise events, the SEL indicator is used when aircraft noise exposure is under consideration, indicating the transient nature of aircraft noise exposure.

It is important to note that all articles in the included literature looked at noise exposure in relation to the *A*-weighting filter, including those where the equivalent noise exposure levels were higher than 50–55 dBA [80]. Although the *A*-weighting filter mimics the human hearing response (i.e., approximate Fletcher–Manson curves [81]), low frequencies are underestimated [80,82], leading to a possible weak association with human perception [83]. In addition, this filter was only designed for levels around 40 dB. In high levels, its performance can be characterized as poor [82], considering the flattening of the Fletcher–Manson curves at these levels. In the included literature, no frequency-based indicators have been used. Although that spectrum-based indicators are known (see ref. [32]), these have never been explored in noise-based studies. The characterization of noise exposures (i.e., indoor and/or outdoor) with respect to noise indicators, based on their spectral characteristics, may lead to different exposure–response curves, especially when considering the transmission of outdoor noise exposure into an indoor environment through a building façade.

#### 4.1.4. Children and Mediating Factors

Except for the direct associations between noise exposure and non-auditory health effects in children, there are parameters mediating these relationships. Such parameters correspond to personal appraisal, coping strategies, age group, and others. Regarding the perception of noise exposure, noise sensitivity is an important mediating personal appraisal factor. This factor is associated with the perception towards various types of noise sources and their characteristics, indicating the degree of reactivity to noise characteristics [84]. In addition, it has been characterized as an independent predictor of annoyance for urban noise [2]. Coping strategies are strongly associated to the degree of exposure, leading to actions of tuning out unwanted noise events [85]. Age group is also an important factor, considering the developmental period of children as well as the development of coping strategies for stress events, such as noise. Gender, genetic characteristics, lifestyle, socioeconomic status, beliefs, and social environment may provide a higher degree of consistency in the analyses of associating with noise exposure and health effects [24]. Approximately 83% of the included literature used mediating factors related to adjusted confounders in the statistical analyses (see Appendix A). In addition, most of the studies included participants criteria, related to type of noise exposure, demographic characteristics, health aspects, and others (see Appendix A).

#### 4.1.5. Matching between Noise Indicators and Noise-Induced Health Effects

Exposure-based studies (i.e., road-traffic and aircraft noise) on psychophysiological [6,7,8,9,10,11], cognitive development [7,9,10,12,13], mental health [14,15], and sleep [16,17] effects have shown consistency with the studies quantifying noise exposure by the $L_{A,eq}$ and the $L_{den}$ indicators. Regarding sleep effects, it is interesting to note that event-based indicators were not identified in children-based studies, even though it is known that noise events are responsible for distorting sleeping activity [86].

Both acute and temporal characteristics of noise exposure, corresponding to the transient nature of noise sources, have shown a significant impact on cognitive performance tasks [63]. More specifically, significant associations between noise exposure and cognitive performance tasks have been revealed with respect to the statistic percentile indicator $L_{A10}$ and the noise event indicator $L_{A,max}$, describing noise events, as well as to the percentile indicator $L_{A90}$ and the equivalent noise indicator $L_{A,eq}$, describing the constant character of the noise exposure. Both the $L_{A,max}$ and the $L_{A10}$ indicator presented significant correlations with cognitive performance tasks in older children. This may indicate that long-term exposure is responsible for the development of coping strategies, filtering out the time-independent patterns of noise exposure. In that way, noise exposure based on events has a dominant effect in the association between noise exposure and cognitive performance task [56]. In contrast to older children, younger children, who have had a shorter exposure, may have less developed coping strategies than chronically exposed children. Therefore, the effects on their performance tasks are possibly associated with the background noise indicator $L_{A90}$ and the equivalent noise indicator $L_{A,eq}$ [56].

Considering the pathways of exposure, the inclusion of both direct (e.g., noise exposure during learning activities) and indirect (e.g., noise exposure in dwellings) exposures affects the performance of children in cognitive tasks. Energy-based indicators, expressing noise exposure in homes, with reference to the indicator $L_{den}$, and in schools, with reference to the indicator $L_{d}$ or the indicator $L_{e}$ or the indicator $L_{n}$, showed a significant effect. The latter evidence is in agreement with previous exposure-based studies [7,13,85]. Finally, the association between the $L_{A,eq}$ indicator in terms of road traffic noise exposure and cognitive functioning-based test results indicated the influence of noise exposure on complex tasks [57], which is in accordance with the study [87].

### 4.2. Quality of Evidence

The defined assessment criteria (see Table 3) have been implemented in terms of the design and outcome in each individual study (see Appendix A). In total, 56% of articles presented a low overall risk of bias, which corresponds to the high quality of evidence. The other 44% corresponds to a high overall risk of bias, indicating the low quality of evidence. From the latter percentage, 6% of the studies presented a high rate of bias with respect to noise exposure assessment, corresponding to non-continuous short-time noise measurements. The remaining 38% corresponds to the other three risk of bias assessment criteria.

## 5. Limitations

In terms of the weaknesses of this study, the search procedure was conducted to a limited number of literature databases, indicating that valuable articles may have not been included to this study due to their inclusion to other literature databases. The screening procedure was conducted by the first author (M.E.T.), meaning that possible biases related to inclusion/exclusion criteria may have been included. Although each screening round was validated in a continuous basis and the included literature was validated by the second author (M.D.), this type of bias can not be neglected.

## 6. Conclusions

Noise exposure in urban environments has been characterized as a health threat for humans. Children, due to their continuous mental and physical development, have been considered as a sensitive group, indicating that noise-induced health effects could be detrimental and possibly irreversible during their adulthood period. Hence, the objective of conducting this systematic literature review was the identification of objective noise indicators associated with non-auditory health effects in children. For the purposes of this review, psychophysiological, cognitive, mental and sleep effects have been considered. From the included literature, it is seen that mostly energetic noise indicators have been used for the association of noise exposure with all the considered non-auditory health effects in children-based studies. Only a small amount of studies (i.e., cognitive performance and mental health effects) used statistical and event-based indicators in combination with energy-based ones. The latter suggests that it is important to further examine what type of noise indicators affect children and to identify possible developed and undeveloped coping repertoires (mechanisms) with respect to noise indicators, indicating an open research topic.

Apart from the currently used indicators, spectral-based noise indicators could have a correlation with non-auditory health effects in children too, and should be subject of future research.

Considering that children spend most of their time at home and at school indoors, indoor noise levels are of greater concern than outdoor noise levels. This underscores the importance of developing and applying outdoor-to-indoor transition correction factors when studying the effects of outdoor noise exposure (e.g., noise maps) on children’s health. Correction factors could be derived either by measurements (e.g., outdoor and indoor measurements) or by modeled data (e.g., propagation-based models) and the application of statistical (learning) methods, including additional parameters related to the outdoor and indoor environment and the building façade characteristics.

Finally, the perceptual part of noise indicators needs to be considered, as noise exposure levels may deviate from the levels that justify *A*-weighting. Therefore, more appropriate weighting following the equal-loudness contours is recommended.

## Figures and Tables

**Figure 1 ijerph-19-15633-f001:**
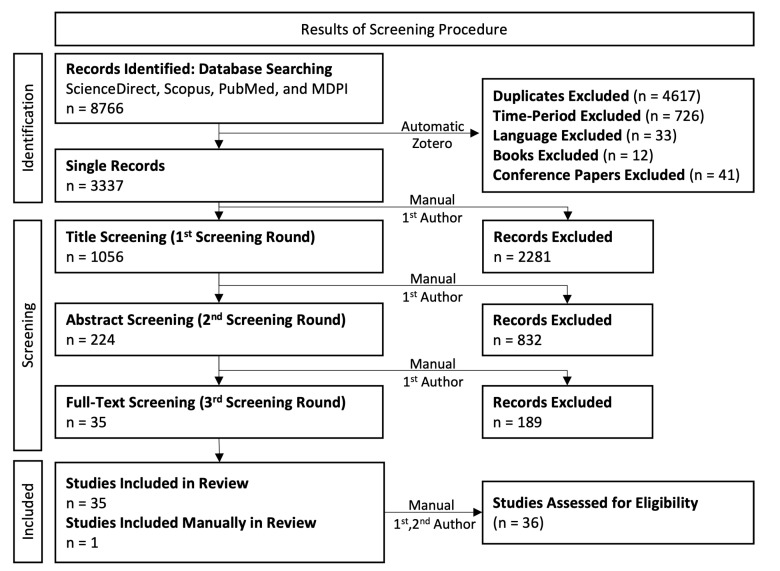
Flow chart of the screening procedure.

**Figure 2 ijerph-19-15633-f002:**
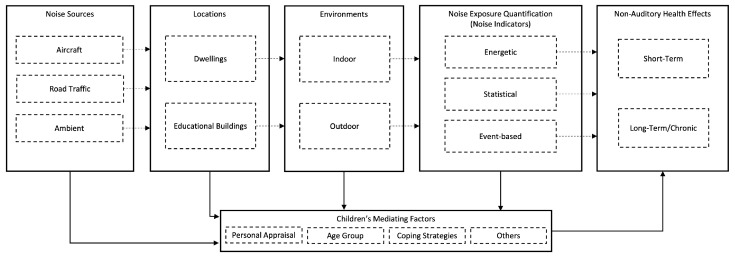
Conceptual framework of the relationship between noise exposure and non-auditory health effects in children.

**Table 1 ijerph-19-15633-t001:** Defined key and search terms.

Key Terms	Search Terms
{Noise Exposure}	“noise indicator(s)”, “noise index(ces)”, “noise metric(s)”, “noise exposure”,
	“environmental noise”, “urban noise”, “industrial noise”, “ambient noise”,
	“traffic noise” and “aircraft noise”
{Health Effects}	annoyance, “cognitive performance”, “cognitive development”, cognition,
	“mental disorder(s)”, “mental health”, psycho[physio]logical, rest[oration],
	sleep, and stress
{Children}	neonatal, fetus, new-born, infant(s), baby(ies), toddler(s), adolescent(s),
	juvenile(s), student(s), and child(ren)

“·”: Search terms as strings, {·}: key term, (·): plural, [·]: combinations.

**Table 2 ijerph-19-15633-t002:** Defined inclusion and exclusion criteria.

	Criterion	Inclusion	Exclusion
(A)	Time Period	2000–2020	<2020
(B)	Language	English	Non-English
(C)	Literature	Research and review articles	Books and conference papers
(D)	Noise Sources	Aircraft, road-traffic, and ambient *	Others
(E)	Noise Indicators	Objective	Subjective
(F)	Locations	Educational and residential buildings	Others
(G)	Environments	Indoor and outdoor	-
(H)	Health Effects	Psychophysiological, cognitive, mental,	Others
		health, and sleep effects	
(I)	Participants	Children (<18 yrs)	Adults (≥18 yrs)
(J)	Others	- Healthy and normal hearing children	- Unclear or absence of noise
		- Multi-exposure and adults-children	exposure in statistical analyses
		studies without combined results	- Reproduction-based noise
			from loudspeakers/headphones

* Dominant noise source road-traffic or aircraft.

**Table 3 ijerph-19-15633-t003:** Scoring protocol used to the assess the risk of bias in studies’ design.

Bias Criterion	Bias Description	Risk of Bias
Participants’ Selection	Participants randomly selected from a known population, AND response rate higher than 60% AND attribution rate less	Low
	than 20% in the follow-up studies.	
	No random sampling OR response rate less than 60% or attributional rate higher than 20%.	High
	No sufficient information to judge.	Unclear
Health Measurement Outcome	Objectively measured outcome OR taken from medical records OR taken from questionnaire using a knowing scale or	Low
	validated assessment method.	
	Self-reported and not assessed using a knowing scale or validated assessment method.	High
	No sufficient information to judge.	Unclear
Noise Exposure Assessment	For long-term exposure: Noise exposure assessment based on measurements for at least one week, OR noise models based on	Low
	actual traffic data OR noise maps verified by noise measurement.	
	For short-term exposure: Noise exposure assessed by continuous measurements during the tasks.	
	For long-term exposure: Measurements less than of a week or not continuous measurements	High
	OR models not verified by measurements or actual traffic data. For short-term exposure:	
	Not continuous measurements during the tasks.	
	No sufficient information to judge.	Unclear
Confounding Factors	All the important confounders have been considered in the analysis.	Low
	Non adjustment of the important confounding factors.	High
	No sufficient information to judge.	Unclear
Overall Bias Rating	All four ratings to bias “Low”.	Low
	One or more “High” or “Unclear”.	High

**Table 4 ijerph-19-15633-t004:** Overview of the collected information extracted from the included literature.

Health Effect	Health Aspect	Parameter	Noise Source	Noise Indicators	Articles (%)
Psycho-physiological	Cardiovascular[37,38,39,40,41]	Systolic and diastolicblood pressure	 	LA,eq1, Ldn, Lden	10
	Neuroendocrine[41,42,43,44]	Cortisol, glucocorticoids,and metabolites	 	LA,eq1, Lden, SEL	8
Cognitive	Development[44,45,46,47,48,49,50,51,52]	Attention, reading, short/long-term memory	 	LA,eq1, SEL	18
	Performance[40,53,54,55,56,57]	Academic achievements,and cognitive functioning	  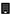	LA,eq1, LAmax,LAmin, Lden,Ld, Le, Ln,LA10, LA90	12
Mental Health	Well-beingDimensions[41,44,45,46,47,49,58,59,60,61,62,63]	Annoyance, motivation,quality of life	  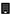	LA,eq1, LAmax,Lden, Ldn,SEL, LA10,LA90, LA99	24
	[40,44,45,49,64,65,66,67,68,69,70]	Psychological morbidity,executive functioning,mental disorders, beha-vioural problems and dis-orders, perceived general,and mental health	  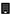	LA,eq1, Lden,Ldn, SEL	22
Sleep	[64,71,72]	Seep quality, waking-upepisodes, sleepiness, dif-ficulties falling sleep		LA,eq1, Len, Ln	6

(·)^1^: Different time periods, 

: Aircraft noise, 

: Road-traffic noise, 
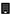
: Ambient noise.

**Table 5 ijerph-19-15633-t005:** Cardiovascular-based studies. SBP and DBP refer to systolic and diastolic blood pressure, respectively. ”+” and ”−” refer to statistical and non-statistical significant associations between outcome and noise indicators, and OBR to the overall bias rating, where L and H correspond to low and high OBR. Finally, dwellings and educational buildings are assigned by 

 and 

, respectively.

Article(Participants, Age Range)	Noise Source(Assessment)	Location(Environment)	Noise Indicator	Outcomes	OBR
SBP	DBP
Wallas et al., (2019) [37] ^†^(n = 2597, 16 yrs)	 C	 O	Lden	+	+	L
Babisch et al., (2009) [38] ^†^(n = 1048, 10–14 yrs)	 M	 O	LA,eq,16h	+	+	H
Belojevic et al., (2008) [39] ^†^ (n = 328, 3–7 yrs)	 M	  O	Lden	+	−	H
Van Kempen et al., (2010) [40] ^†^(n = 2844, 9–11 yrs)	  C	 O	LA,eq,16h	+	+	L
Lercher et al., (2013) [41] ^†^(n = 115, mean 10.1 yrs)	 C	 O	Ldn	−	−	L

†: Cross-sectional study, *C*: Calculations, *M*: Measurements, *O*: Outdoor.

**Table 6 ijerph-19-15633-t006:** Neuroendocrine-based studies. CL and CML refer to cortisol and its metabolite levels as well as GL and GML to glucocorticoid and its metabolite levels. For an explanation of symbols and notations, see the caption of Table 5.

Article(Participants, Age Range)	Noise Source(Assessment)	Location(Environment)	Noise Indicator	Outcomes	OBR
CL	CML	GL	GML
Wallas et al., (2018) [42] ^‡^(n = 1751, 16 yrs)	 C	 O	Lden	−				L
Cantuaria et al., (2018) [43] ″(n = 165, 5 wks)	 C	 O	Lden	−	+	−	+	L
Haines et al., (2001) [44] ^†^(n = 340, 8–11 yrs)	 M,C	 I	SEL	−				L
Lercher et al., (2013) [41] ^†^ (n = 115, mean 10.1 yrs)	 C	 O	Ldn	+	+			L

†: Cross-sectional study, ‡: Follow-up study, ″: Population-based study, *C*: Calculations, *M*: Measurements, *O*: Outdoor, *I*: Indoor.

**Table 7 ijerph-19-15633-t007:** Cognitive development-based studies. SA corresponds to sustained attention, RD to reading, WM to working memory, RGNM to recognition memory, RCLM to recall memory, and PM to prospective memory. For explanation of symbols and notations, see caption of Table 5.

Article(Participants, Age Range)	Noise Source(Assessment)	Location(Environment)	Noise Indicator	Outcomes	OBR
SA	RD	WM	RGNM	RCLM	PM
Clark et al., (2013) [45] ^‡^(n = 461, 15–17 yrs)	 C	 O	LA,eq,16h		−					L
Klatte et al., (2017) [46] ^†^(n = 1243, 7–10 yrs)	 C	  O	LA,eq,6h,LA,eq,12h		+					H
Stansfeld et al., (2005) [47] ^†^(n = 2844, 9–10 yrs)	  M,C	 O	LA,eq,16h	−	+	−	+	−	−	L
Seabi et al., (2015) [48] ^‡^(n = 650/178, 9–15/10–16 yrs)	 M	 O	LA,eq,2.5h		−					H
Haines et al., (2001) [44] ^†^(n = 340, 8–11 yrs)	 M,C	 I	LA,eq,16h		+	−	+	−		L
Haines et al., (2001) [49] ^†^(n = 275, 8–11 yrs)	 M,C	 I	SEL	+	+					L
Clark et al., (2006) [50] ^††^(n = 2844, 9–10 yrs)	  M,C	  O	LA,eq,16h		+					L
Matsui et al., (2004) [51] ^†^(n = 236, 8–9 yrs)	 C	 O	LA,eq,16h	−	−		−	+		L
Matheson et al., (2010) [52] ^†^(n = 2844, 8–12 yrs)	  C	 C	LA,eq,16h				+	+	−	L

†: Cross-sectional study, ‡: Follow-up study, ††: Longitudinal study, *C*: Calculations, *M*: Measurements, *O*: Outdoor, *I*: Indoor.

**Table 8 ijerph-19-15633-t008:** Cognitive performance-based studies. AA corresponds to academic achievements, LNG to language, MTH to mathematics, SCN to science, and CF to cognitive functioning. For an explanation of symbols and notations, see the caption of Table 5.

Article(Participants, Age Range)	Noise Source(Assessment)	Location(Environment)	Noise Indicator	Outcomes	OBR
AA	LNG	MTH	SCN	CF
Xie et al., (2011) [53] * (n = n.sp., 12–13 yrs)	 C	 O	LS,LS,max,LS,min LS,LS,10,LS,90	−					H
Pujol et al., (2014) [54] ^†^ (n = 587, 8–9 yrs)	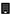 C	  O	Ld,Le,Ln, Lden		+	+			L
Van Kempen et al., (2010) [40] ^†^(n = 2844, 9–11 yrs)	  C	 O	LA,eq,16h					+	L
Haines et al., (2002) [55] †(n = 11,000, 11 yrs)	 C	 O	LA,eq,16h		+	+	−		H
Shield et al., (2008) [56] * (n = n.sp./24, 7, 11/6–7, 10–11 yrs)	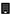 M	 O,I	LA,eq,5m,LA,10,5m, LA,90,5m,LA,max,5m		+	+	+		H
Van Kempen et al., (2010) [57] ^†^(n = 553, 9–11 yrs)	  C	  O	LA,eq,16h					+	L

†: Cross-sectional study, *: Case study, *C*: Calculations, *M*: Measurements, *O*: Outdoor, *I*: Indoor.

**Table 9 ijerph-19-15633-t009:** Well-being dimensions-based studies. ANN corresponds to annoyance, MOT to motivation, and QOL to quality of life. For explanation of symbols and notations, see caption of Table 5.

Article(Participants, Age Range)	Noise Source(Assessment)	Location(Environment)	Noise Indicator	Outcomes	OBR
ANN	MOT	QOL
Clark et al., (2013) [45] ^‡^(n = 461, 15–17 yrs)	 C	 O	LA,eq,16h	+			L
Klatte et al., (2017) [46] ^†^(n = 1243, 7–10 yrs)	 C	  O	LA,eq,6h,LA,eq,12h	+		+	H
Ali (2013) [58] ′(n = 300, 13–15 yrs)	 M	 I	LA,eq,20m	+			H
Stansfeld et al., (2005) [47] ^†^(n = 2844, 9–10 yrs)	  M,C	 O	LA,eq,16h	+			L
Silva et al., (2016) [59] *^o^*(n = 213, 8–11 yrs)	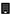 M	 O,I	LA,eq,30m,LA,95,30m	+			H
Van Kempen et al., (2009) [60] ^†^(n = 2844, 9–11 yrs)	  C	  O	LA,eq,16h	+			L
Haines et al., (2001) [44] ^†^(n = 340, 8–11 yrs)	 M,C	 I	LA,eq,16h	+	−		H
Haines et al., (2001) [49] ^†^(n = 275, 8–11 yrs)	 M,C	 I	SEL	+			L
Minichilli et al., (2018) [61] ^†^(n = 521, 11–17 yrs)	 M	 O,I	LA,eq,ext,30m LA,eq,int,30m	+			H
Birk et al., (2011) [62] ^†^(n = 951, 10 yrs)	 C	 O	Lden	+			H
Dockrell et al., (2004) [63] ^†^(n = 2063, 6–7, 10–11 yrs)	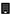 M	 O	LA,eq,5m,LA,max, LA10,LA90,LA99	+			H
Lercher et al., (2013) [41] ^†^(n = 115, mean 10.1 yrs)	 C	 O	Ldn			+	H

†: Cross-sectional study, *o*: In-situ study, ‡: follow-up study, ′: Social survey, *C*: Calculations, *M*: Measurements, *O*: Outdoor, *I*: Indoor.

**Table 10 ijerph-19-15633-t010:** Mental health effects-based studies. SDQ corresponds to the total score of the psychological morbidity SDQ test. ES corresponds to emotional symptoms, CP to conduct problems, HI to hyperactivity/inattention, PP to peer problems, PSB to pro-social behavior problems (SDQ sub-scales). AD to anxiety/depression, PH to perceived general and mental health, BP to behavioral problems, and EF to executive functioning skills. For explanation of symbols and notations, see caption of Table 5.

Article(Participants, Age Range)	Noise Source(Assessment)	Location(Environment)	Noise Indicator	Outcomes	OBR
ES	CP	HI	PP	PSB	SDQ	AD	PH	BP	EF
Clark et al., (2013) [45] ^‡^(n = 461, 15–17 yrs)	 C	 O	LA,eq,16h	−	−	−			−					L
Tiesler et al., (2013) [64] ^†^(n = 872, 10 yrs)	 C	 C	Lden	+	−	+	−	−	−					H
Stansfeld et al., (2009) [65] ^†^(n = 2844, 9–10 yrs)	  M,C	 O	LA,eq,16h	−	+	+	−	−	−					L
Van Kempen et al., (2010) [40] ^†^(n = 2844, 9–10 yrs)	  C	 O	LA,eq,16h								−			L
Hjortebjerg et al., (2016) [66] ^|^(n = 46,940, 9–11 yrs)	 C	 O	Lden	−	−	+	−	−	−					H
Haines et al., (2001) [44] ^†^(n = 340, 8–11 yrs)	 M,C	 I	LA,eq,16h	−	−	−	−	−	−	−				L
Haines et al., (2001) [49] ^†^(n = 275, 8–11 yrs)	 M,C	 I	SEL							−	+			L
Weyde et al., (2017) [67] ^††^(n = 1934/1384, 0/3–8 yrs)	 C	 O	Lden			+								L
Lim et al., (2018) [68] ^†^(n = 918, 9–14 yrs)	 C	 O	Ldn									+		H
Crombie et al., (2011) [69] ^†^(n = 1900, 9–10 yrs)	  C	 O	LA,eq,16h	−	+	+			−					L
Belojevic et al., (2012) [70] ^†^(n = 311, 7–11 yrs)	 M	  O	LA,eq,24h										+	H

†: Cross-sectional study, ‡: Follow-up study, ††: Longitudinal study, |: Cohort study, *C*: Calculations, *M*: Measurements, *O*: Outdoor, *I*: Indoor.

**Table 11 ijerph-19-15633-t011:** Sleep-based studies. DFS corresponds to difficulties fall sleeping, SP to sleep problems, SQ to sleep quality, WE to waking episodes, and FA to feeling alertness. For explanation of symbols and notations, see the caption of Table 5.

Article(Participants, Age Range)	Noise Source(Assessment)	Location(Environment)	Noise Indicator	Outcomes	OBR
DFS	SP	SQ	WE	FA
Tiesler et al., (2013) [64] ^†^(n = 872, 10 yrs)	 C	 O	Ln	+	+				H
Ohrstrom et al., (2006) [71] ^†^(n = 160, 9–12 yrs)	 C	 O	LA,eq,24h	−		+	+	+	H
Weyde et al., (2017) [72] ^†^(n = 2665, 7 yrs)	 C	 O	Len			−			H

†: Cross-sectional study, *C*: Calculations, *O*: Outdoors.

## Data Availability

Not applicable.

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
