# Peer review of "Noise Indicators Relating to Non-Auditory Health Effects in Children—A Systematic Literature Review"

_ijerph, 2022, doi:10.3390/ijerph192315633_

Round 1

Reviewer 1 Report

Overall, great work! A few suggestions, large and small:

* It would be much better if there were a section of the discussion that discussed the findings of source of the noise pollution. Overall, from all of the papers studied, what are the percentages of noise pollution attributed to vehicular traffic, airplanes and others? Importantly, what about the noise from trains? Construction? Publicly blasted music? 

* Further, it would be much more helpful if the conclusion included suggestions/recommendations for possible solutions to the problem. 

* The search term was "foetus" -- what about the search term "fetus," the preferred spelling in much of the English-language speaking world? 

* In section 3.1: Clarify that "first author" and "second author" are referring to the authors of the submitted manuscript and not to the authors of the papers that were studied. 

Author Response

Dear Reviewer,

Firstly, we would like to thank you for your suggestions and comments. We are really grateful for your valuable inputs. Below, you will find the list of our response (i.e., blue color) to your comments (i.e., black color). Please note that page, figure and table numbers refer to the new pdf manuscript document.

Thank you!

Best Regards

Michail Evangelos Terzakis

Maud Dohmen

Irene van Kamp

Maarten Hornikx 

1. It would be much better if there were a section of the discussion that discussed the findings of source of the noise pollution. Overall, from all of the papers studied, what are the percentages of noise pollution attributed to vehicular traffic, airplanes and others? Importantly, what about the noise from trains? Construction? Publicly blasted music?

Three main sources are under consideration: aircraft, road-traffic, and ambient (where the dominant source is usually road or aircraft traffic) noise. To make it even more clear, in Table 2, a line was added corresponding to the “(D)” inclusion and exclusion criteria. For your information, this line does not add any new information to the defined criteria. This can be seen in Table 1, where only search terms referred to these types on noise source and their exposure are included. This addition in Table 2 pinpoints further the focus on these three types of noise source and exposures.

Because children spend most of their times at on schools and in dwellings, the exposure to construction-based noise pollution could be characterized as a “special case”, which is out of the main noise sources considered in this review. However, this could be definitely kept in our agenda for investigation in the future. Similarly, music, as an exposure, depends on the taste of children, indicating the presence of either positive or negative characteristics. Therefore, it was decided to not focus on it as a primary noise source. It was also decided to not include rail noise, keeping our interest to the aforementioned main source of noise exposure in mind.

Finally, the percentages of noise sources contributed to each article have been included into the “Noise Sources” subsection (lines 339-342).

2. Further, it would be much more helpful if the conclusion included suggestions/recommendations for possible solutions to the problem. 

In the conclusion section, both suggestions and recommendations have been included (lines 527-555).

3. The search term was "foetus" -- what about the search term "fetus," the preferred spelling in much of the English-language speaking world? 

The search term that was used is “fetus” as you mentioned in your comment and not “foetus”. This was a typo from the side of the first author.

4. In section 3.1: Clarify that "first author" and "second author" are referring to the authors of the submitted manuscript and not to the authors of the papers that were studied. 

The clarification of the authors (of this paper) in the parts where they are referred was assigned using their initials in parentheses aside such "first author (M.E.T)" and "second author (M.D)".

Reviewer 2 Report

This review explored noise indicators associated with non-auditory health effects in children concerning the type of noise source, the locations of noise assessment, the exposure locations, and their environments with the influence of children mediating factors. The review was described comprehensively. But certain parts need clarification:

1. What do the authors mean by “studies included manually in review” in the PRISMA flow chart diagram?

2. The authors repeatedly explained the screening procedure in the methods section (line 136-158) and the results section (line 176-189). I suggest combining the procedure and including it in the methods section only.

3. I recommend inserting the citations in Table 4.

4. For Table 4 to Table 11,” the authors should interpret and explain briefly about the tables instead of only presenting the tables (e.g., highlight the most important, etc.).

5. The authors should interpret Figure 2. It perhaps means, noise sources correlate with locations, locations correlate with the environment, the environment is associated with noise indicators, and noise indicators are associated with non-auditory health effect mediated by children mediating factors. But what was understood from the texts was that noise exposures (noise sources, location, and environments) measured by noise indicators had an association with non-auditory health effects and this association was mediated by children mediating factors as illustrated below. Please make it clear.

6. There were results statements (paragraphs 2 to 6) that were included in the conclusion section. It should be moved to the results section. The conclusion should add recommendations related to the study.

Author Response

Dear Reviewer,

Firstly, we would like to thank you for your suggestions and comments. We are really grateful for your valuable inputs. Below, you will find the list of our response (i.e., blue color) to your comments (i.e., black color). Please note that page, figure and table numbers refer to the new pdf manuscript document.

Thank you!

Best Regards

Michail Evangelos Terzakis

Maud Dohmen

Irene van Kamp

Maarten Hornikx

1. What do the authors mean by “studies included manually in review” in the PRISMA flow chart diagram?

Although that the searching procedure was conducted in a specific number of search engines, there are possible valuable articles which have not been shown up. In terms of the included article, this was in our knowledge due to our interest in noise indicators and health effects. Also, it was not derived from any of the used search engines. Therefore, it was decided, this to be included in a manual way without violating the systematic searching approach. This information is presented in lines 160-165.

2. The authors repeatedly explained the screening procedure in the methods section (line 136-158) and the results section (line 176-189). I suggest combining the procedure and including it in the methods section only.

To avoid repetition, the referred two parts where combined together in the procedure section, in the subsection “Screening procedure” (i.e., lines 135-165).

3. I recommend inserting the citations in Table 4.

The citations were inserted in the Table 4, as it was recommended, under each health aspect or effect.

4. For Table 4 to Table 11,” the authors should interpret and explain briefly about the tables instead of only presenting the tables (e.g., highlight the most important, etc.).

The interpretation of the correlations between noise exposure, quantified via (objective) noise indicators and non-auditory health effects in children could not be included in this review since their interpretation is based on various mechanisms (i.e., very briefly described in the introduction). Also, factors such as the age group of the children and the study design of each article and investigated parameter, increase the difficulty of providing an interpretation. These indicate the main difficulties of combining these results and extract their interpretations. Also, the investigation of various mechanisms with respect to health effects is in our agenda for future tasks.

5. The authors should interpret Figure 2. It perhaps means, noise sources correlate with locations, locations correlate with the environment, the environment is associated with noise indicators, and noise indicators are associated with non-auditory health effect mediated by children mediating factors. But what was understood from the texts was that noise exposures (noise sources, location, and environments) measured by noise indicators had an association with non-auditory health effects and this association was mediated by children mediating factors as illustrated below. Please make it clear.

This part was included below the Figure 2. For making it clearer, this part was re-written with respect to your comments (lines 319-332).

6. There were results statements (paragraphs 2 to 6) that were included in the conclusion section. It should be moved to the results section. The conclusion should add recommendations related to the study.

The result statements were removed, and recommendations and suggestions related to the study have been added. These can be seen in the lines 527-555.

Round 2

Reviewer 1 Report

Thank you for a much-improved manuscript!